# Consistency is Key:
# On Data-Efficient Modality Transfer in Speech Translation

**Hojin Lee**
Kakao Brain
lambda.x@kakaobrain.com

**Changmin Lee**
Kakao Brain
louie.cm@kakaobrain.com

**Seung-won Hwang\***
Seoul National University
seungwonh@snu.ac.kr

## Abstract

End-to-end approaches have shown promising results for speech translation (ST), but they suffer from its data scarcity compared to machine translation (MT). To address this, *progressive training* has become a common practice, of using external MT data during the fine-tuning phase. Despite of its prevalence and computational overhead, its validity is not extensively corroborated yet. This paper conducts an empirical investigation and finds that progressive training is ineffective. We identify learning-forgetting trade-off as a critical obstacle, then hypothesize and verify that consistency learning (CL) breaks the dilemma of learning-forgetting. The proposed method, which combines knowledge distillation (KD) and CL, outperforms the previous methods on MuST-C dataset (Di Gangi et al., 2019) even without additional data, and our proposed consistency-informed KD achieves additional improvements against KD+CL. Code and models are availble at https://github.com/hjlee1371/consistency-s2tt.

## 1 Introduction

While traditional speech-to-text translation (ST) systems are built by pipelining automatic speech recognition (ASR) and machine translation (MT), end-to-end (E2E) approach recently emerges as a promising direction to ameliorate error propagation and model complexity problems (Anastasopoulos et al., 2022; Bentivogli et al., 2021). However, E2E ST models encounter data scarcity due to the need for cross-modal annotations, which are less abundant compared to datasets used in related tasks such as machine translation.

Our goal is to enable effective cross-modal transfer from machine translation (MT) models, which have ample training data, to ST models with limited data. In pursuit of this goal, we investigate

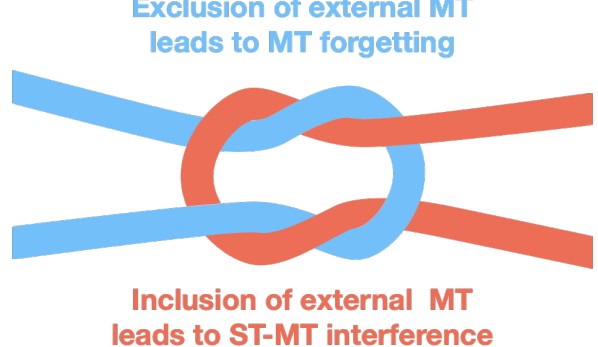

Figure 1: Schematic diagram[1] illustrating the intrinsic tension in adding MT data for ST training. When external MT data is added during finetuning (in red), the model experiences ST-MT interference. Conversely, when less external MT data is incorporated (in blue), the model tends to forget MT knowledge, resulting in suboptimal cross-modal transfer.

the widely used *progressive training* technique in ST (Tang et al., 2021b,a; Ye et al., 2022; Tang et al., 2022; Ye et al., 2021), where external MT data is continuously integrated during the fine-tuning phase. However, our evaluation brings forth surprising results, as we find that progressive training is inadequate and leads to suboptimal outcomes.

This inadequacy is due to the dilemma in adding MT data for ST training. Using external MT data may incur interference between the two tasks, but having less external MT data leads to forgetting.

To break the knot in Figure 1, we first shed light on the overlooked relationship between consistency and forgetting. In addition, we introduce a novel approach called *consistency*-informed knowledge distillation (cKD). Our findings and proposed methods are thoroughly evaluated on the MuST-C benchmark, encompassing various language pairs. The results demonstrate the superior performance and enhanced data efficiency of our approach compared to previous methods.

Our contributions are as follows.

---

\*Corresponding author
[1] Image from PresentationGO

- We reevaluate the validity of the widespread use of progressive training, and find that it is ineffective.
- We find that consistency learning (CL) remedies the catastrophic forgetting problem, and a simple combination of knowledge distillation (KD) and CL achieves promising ST BLEU scores with a simple design choice and data-efficient training.
- We further push the limit of KD+CL through proposing consistency-informed KD (cKD), which utilizes token-level consistency for adaptive weighting of KD.

## 2 Motivation: Is Progressive Training Effective?

**Problem statement and Baseline**   To motivate, we define our problem and describe a *progressive learning* baseline by Ye et al. (2021), that we use as strong baseline throughout the paper.

The speech translation (ST) corpus, denoted as $\mathcal{D}_{ST} = \{(\mathbf{s}, \mathbf{x}, \mathbf{y})\}$, consists of $\mathbf{s}$ (source language speech), $\mathbf{x}$ (source language text or *transcription*), and $\mathbf{y}$ (target language text or *translation*).

Due to the scarcity of speech translation datasets, it is common practice to train ST models jointly with MT or ASR subtasks (Tang et al., 2021a,b; Ye et al., 2021). Similarly, in our approach, we train our model jointly on ST and MT using multitask cross-entropy losses, denoted as $\mathcal{L}_{CE}(\mathbf{s}, \mathbf{y}, \theta)$ and $\mathcal{L}_{CE}(\mathbf{x}, \mathbf{y}, \theta)$.

In *progressive training*, the MT training is continued during the fine-tuning phase using external data source $\mathcal{D}_{ext} = \{(\mathbf{x}, \mathbf{y})\}$. Specifically, at each epoch, $\mathcal{D}_{ext}$ is randomly downsampled to $\mathcal{D}'_{ext}$, and during training, ST triplets $(\mathbf{s}, \mathbf{x}, \mathbf{y})$ or MT pairs $(\mathbf{x}, \mathbf{y})$ are sampled from the union of $\mathcal{D}_{ST}$ and $\mathcal{D}'_{ext}$.

In addition to joint training, we incorporate MT-to-ST online knowledge distillation (KD) proposed by Tang et al. (2021a). For the data triplet $(\mathbf{s}, \mathbf{x}, \mathbf{y})$, the KD loss is computed as:

$$\mathcal{L}_{KD} = -\sum_{i=1}^{|\mathbf{y}|} \sum_{j=1}^{|V|} P(y_i = v_j | \mathbf{y}_{i<}, \mathbf{x}; \theta) \tag{1}$$
$$\log P(y_i = v_j | \mathbf{y}_{i<}, \mathbf{s}; \theta)$$

where $v_j$ corresponds to the $j$-th token of the vocabulary. This encourages the ST "student" to learn more fine-grained information from the MT "teacher". When combined with baseline systems,

$\mathcal{L}_{KD}$ is weighted by $\alpha_{KD}$ and added to the final loss.

**Progressive training is ineffective**   Despite its widespread adoption, we propose that the efficacy of progressive training has been accepted without sufficient empirical evidence and followings highlight our empirical findings against common beliefs. From Table 1, we can see that progressive training **does not improve** the ST performance, despite expensive computational overhead, contrary to the popular belief. For deeper understanding, we also evaluate its MT performance throughout the training using *transcription-translation* pair of ST triplet. As depicted in Figure 2, we observe catastrophic forgetting in MT performance when we train ST model without $\mathcal{D}_{ext}$, while it preserves its MT knowledge by training with $\mathcal{D}_{ext}$.

Based on our observation of catastrophic forgetting, it might be expected that cross-modal KD would benefit from progressive training, as augmented MT data provides a more reliable teacher signal. However, the inclusion of an extra KD objective in Table 1 does not yield significant improvements. This raises an important question: why does addressing catastrophic forgetting not lead to improved ST performance? It appears that while $\mathcal{D}_{ext}$ mitigates forgetting, it also diverts the model's focus from ST, highlighting the inherent tension between learning ST and avoiding MT forgetting.

| Lang. | Models | $\mathcal{D}_{ST}$ | $+\mathcal{D}_{ext}$ | $p$-value |
|-------|--------|------|------|---------|
| De | Baseline | 28.07 | 28.07 | 0.4145 |
| Es | Baseline | 31.35 | 31.30 | 0.2686 |
| Fr | Baseline | 37.94 | 38.04 | 0.1600 |
| De | Base+KD | 28.18 | 28.23 | 0.2456 |
| Es | Base+KD | 31.52 | 31.37 | 0.1186 |
| Fr | Base+KD | 38.28 | 38.33 | 0.2541 |

Table 1: ST BLEU results with the baseline and KD training for various translation directions. $\mathcal{D}_{ext}$ is additional MT data from an external source(e.g. WMT) other than ST triplet datasets.

## 3 Proposed: Effective Cross-modal Transfer with Consistency Learning and Consistency-informed KD

**Continual Learning View of Speech Translation** ST learning can be viewed from a continual learning perspective, where knowledge is continuously

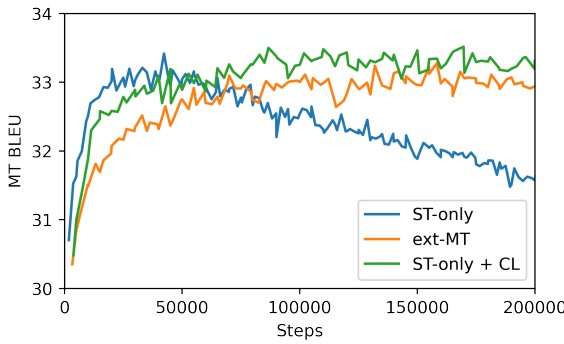

Figure 2: Evolution of MT BLEU with *transcription-translation* pairs from MuST-C En-De dev set.

accumulated through pre-trained MT, MT-, and ST-finetuning. From this standpoint, progressive training can be interpreted as the generalization of *rehearsal* (Robins, 1995). However, as highlighted in studies on continual learning (Verwimp et al., 2021) and multitask learning (Lin et al., 2019; Sener and Koltun, 2018), bare replay of previous tasks is not the answer to the stability-plasticity dilemma: it only offers a suboptimal balance between MT and ST. Therefore, we turn our focus to regularization-based methods such as elastic weight consolidation (EWC) and its variants (Kirkpatrick et al., 2017; Chaudhry et al., 2018; Schwarz et al., 2018; Thompson et al., 2019).

Their main idea is to restrict the gradient of new tasks within the low-curvature "valleys" of the loss landscape of previous task. While traditional EWC approximates the Hessian as the diagonals of the Fisher information matrix $\sum_i F_{ii}(\theta_i - \theta_i^*)^2$, we can remove diagonal approximation from EWC through well-known relation between KL divergence and Fisher as

$$\mathcal{D}_{KL}(p_{\theta^*}||p_\theta) \approx E_z[\log p_{\theta^*} - \log p_\theta]$$
$$= \frac{1}{2}(\theta - \theta^*)^T F^*(\theta - \theta^*) \quad (2)$$

While $\theta^*$ is fixed parameter obtained from prior tasks in original EWC, we can interpret it as current parameter and $\theta = \theta^* + \Delta\theta$ as the parameter of submodel induced by arbitrary perturbations. It recalls recently proposed CL losses such as R-Drop (Liang et al., 2021), and provokes the question that whether the CL framework can ameliorate the intrinsic dilemma of ST. We thus adopt R-drop as our CL framework, and weighted loss $\alpha_{CL}\mathcal{L}_{CL}$ is added to the final loss.

**Consistency-informed KD** If the concept of consistency is indeed crucial for preventing MT forget-

ting, as we hypothesized, we can expect substantial improvements by incorporating cross-modal KD compared to not using it. To achieve further enhancements, we can go beyond the basic combination of KD and CL. In particular, we introduce a novel loss called consistency-informed KD (cKD), which leverages more detailed information on consistency, as outlined below.

$$\mathcal{L}_{cKD} = -\sum_{i=1}^{|\mathbf{y}|}\sum_{j=1}^{|V|} e^{-c_{ij}^{MT}} P(y_i = v_j|\mathbf{y}_{i<}, \mathbf{x}; \theta)$$
$$\log P(y_i = v_j|\mathbf{y}_{i<}, \mathbf{s}; \theta)$$

It augments vanilla KD with token-level weighting based on MT consistency matrix $c^{MT}$. Concretely, $c_{ij}^{MT}$ represents the bidirectional KL divergence between two forward pass probabilities for $i$-th token and $j$-th vocabulary: $P_1(y_i = v_j|\mathbf{y}_{i<}, \mathbf{x})$ and $P_2(y_i = v_j|\mathbf{y}_{i<}, \mathbf{x})$. Intuitively, cKD can be understood as ignoring inconsistent MT teacher probabilities at token-level.

## 4 Results and Analysis

**Consistency learning remedies catastrophic forgetting** We begin by examining the hypothesis that CL data-efficiently remedies catastrophic forgetting. As illustrated in Figure 2, CL demonstrates a remarkable ability to retain MT knowledge even in the absence of additional MT data, thereby confirming our hypothesis. The final ST BLEU scores, presented in Table 3, further support these findings. Surprisingly, progressive training consistently underperforms in all language directions. This suggests that progressive training becomes redundant in the presence of CL, as it loses the benefits of preserving MT knowledge while still diverting the models' attention away from ST.

**CL provides a more reliable teacher for KD** Thanks to our data-efficient solution for catastrophic forgetting, we can confidently predict larger gains from knowledge distillation (KD), as explained in 3. To empirically demonstrate this, we train our ST model using a simple combination of KD and CL. The results in Table 3 clearly show that this approach leads to greater improvements in all language directions, surpassing the performance of progressive training by a significant margin.

Furthermore, we observe that the performance gain from KD is more pronounced when combined with CL compared to KD alone (+0.29 BLEU with

| Models | Joint PT | FT Data | | Languages | | | Avg. |
|---|---|---|---|---|---|---|---|
| | | ST | MT | De | Es | Fr | |
| TaskAware[‡](Indurthi et al., 2021) | ✓ | ✓ | - | 28.88 | - | - | - |
| SpeechT5(Ao et al., 2022) | ✓ | ✓ | - | 25.18 | - | 35.30 | - |
| STPT[‡](Tang et al., 2022) | ✓ | ✓ | ✓ | 29.2[§] | 33.1 | 39.7 | 34.0 |
| SpeechUT[‡](Zhang et al., 2022) | ✓ | ✓ | - | **30.1** | **33.6** | **41.4** | **35.0** |
| JT-S-MT(Tang et al., 2021a) | - | ✓ | ✓ | 26.8 | 31.0 | 37.4 | 31.7 |
| XSTNet(Ye et al., 2021) | - | ✓ | ✓ | 27.8[†] | 30.8 | 38.0 | 32.2 |
| Chimera(Han et al., 2021) | - | ✓ | - | 27.1[†] | 30.6 | 35.6 | 31.1 |
| SATE(Xu et al., 2021) | - | ✓ | - | 28.1[†] | - | - | - |
| STEMM(Fang et al., 2022) | - | ✓ | - | 28.7 | 31.0 | 37.4 | 32.4 |
| WACO(Ouyang et al., 2022) | - | ✓ | - | 28.1 | 32.0 | 38.1 | 32.7 |
| AdaTrans(Zeng et al., 2022) | - | ✓ | - | 28.7 | - | 38.7 | - |
| ConST(Ye et al., 2022) | - | ✓ | ✓ | 28.3 | 32.0 | 38.3 | 32.8 |
| Ours(Base+KD+CL) | - | ✓ | - | 29.08 | 32.13 | 39.47 | 33.56 |
| Ours(Base+cKD+CL) | - | ✓ | - | **29.27**[**] | **32.32**[*] | **39.51** | **33.70** |

Table 2: ST BLEU scores on MuST-C `tst-COMMON` for various methods. † use OpenSubtitles (Lison and Tiedemann, 2016) for $\mathcal{D}_{ext}$ and ‡ use additional data augmentation. § is from corresponding github implementations, not from the papers. * and ** indicates statistically significant differences between (Base+KD+CL) and (Base+cKD+CL) ($p < 0.1$ and $p < 0.05$).

| Lang. | Models | $\mathcal{D}_{ST}$ | $+\mathcal{D}_{ext}$ | $p$-value |
|---|---|---|---|---|
| De*** | Base+CL | **28.94** | 28.57 | 0.0043 |
| Es*** | Base+CL | **31.83** | 31.18 | 0.0001 |
| Fr** | Base+CL | **39.05** | 38.79 | 0.0253 |
| De** | +KD+CL | **29.08** | 28.82 | 0.0240 |
| Es*** | +KD+CL | **32.13** | 31.43 | 0.0001 |
| Fr*** | +KD+CL | **39.47** | 38.96 | 0.0001 |

Table 3: ST BLEU results with CL training for various translation directions. *** indicates statistically significant differences between $\mathcal{D}_{ST}$ and $\mathcal{D}_{ST} + \mathcal{D}_{ext}$ ($p < 0.01$).

CL vs. +0.21 BLEU without CL). This suggests that the improvements achieved through CL are not limited to intra-modal regularization, but rather have a broader cross-modal impact. Thus, we can attribute the enhanced performance to a better MT teacher originating from the non-forgetting effect of CL.

**Additional improvement from cKD and comparison of methods** From Table 2, it is evident that our straightforward combination of KD and CL outperforms the majority of previous methods, even with minimal FT data. Moreover, our proposed cKD method achieves additional significant improvements. While we included large-scale joint pretraining (JPT) methods in the comparison, it is important to note that these methods require signif-

icantly more data, training complexity, and computational resources[2]. Despite this, our method performs on par with most of the JPT approaches, indicating ample opportunity for further research in developing lightweight strategies for patching modality-specific pretrained models.

**Simple KD+CL is comparable to well-chosen** $\mathcal{D}_{ext}$ Some previous works have observed that introducing the spoken domain $\mathcal{D}_{ext}$, such as OpenSubtitles (Lison and Tiedemann, 2016), improves ST BLEU (Ye et al., 2021; Han et al., 2021). To compare our data-efficient method with more competitive models, we also conducted intensive experiments for En-De using OpenSubtitles as our new $\mathcal{D}_{ext}$ during finetuning. Without CL, models trained with OpenSubtitles achieve higher BLEU scores, which aligns with previous works and demonstrates the importance of domain knowledge. However, with CL, training with a well-chosen external MT becomes worthless.

Considering the difficulties and high computation costs of determining the best $\mathcal{D}_{ext}$, our suggested approach provides a more practical and efficient way of training ST models. Further detailed analysis of the relationship between CL and forgetting can be found in the appendix E.

---

[2]For detailed information, refer to appendix D

| Models | $\mathcal{D}_{\text{ST}}$ | +OpenSubtitles | $p$-value |
|--------|------|-------|-------|
| Base*** | 28.07 | **28.54** | 0.0001 |
| +KD*** | 28.18 | **28.66** | 0.0001 |
| +CL | 28.94 | 28.90 | 0.2942 |
| +KD+CL | 29.08 | 28.91 | 0.1036 |

Table 4: ST BLEU results with various training configurations for En-De, after substituting WMT with OpenSubtitles. *** indicates statistically significant differences between $\mathcal{D}_{\text{ST}}$ and $\mathcal{D}_{\text{ST}} + \mathcal{D}_{\text{ext}}$ ($p < 0.01$).

## 5   Conclusion

In this paper, we conduct thorough experiments to reexamine the effectiveness and efficiency of progressive training. We identify the key challenge of balancing ST and MT tasks and discover that, when KD and CL are combined with a balance, adding data plays a deterimental role and thus can be omitted for higher data efficiency. Our findings lead us to propose cKD, which dynamically utilizes intra-modal consistency for cross-modal KD. Our experiments demonstrate the effectiveness of cKD in terms of both performance and data efficiency. We also provide additional analysis to support our findings.

## 6   Limitations

While our method provides a simple and data-efficient way of training, the convergence speed of a single model is still slow, as previously reported in Liang et al. (2021). Although Beyer et al. (2022) recently studied the advantage of lengthy training schedules in knowledge distillation, KD with a faster convergence remains as an open question.

Additionally, while our codes and data are sourced from publicly available resources, our pre-trained MT checkpoints are not publicly accessible. Although this choice was necessary to ensure a fair comparison with previous studies, leveraging public MT checkpoints could potentially enable more efficient training strategies, especially for low-resource languages. Notably, the availability of extensive multilingual MT checkpoints, such as those introduced by Fan et al. (2021); Ma et al. (2021), presents an opportunity for enhanced training approaches.

Furthermore, the translation directions we considered are quite limited to Indo-European languages (German, Spanish, and French), though this is partially attributed to the scarcity of non-Indo-European benchmarks.

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

## A   Data Statistics

|    |      | En-De | En-Es | En-Fr |
|----|------|-------|-------|-------|
| ST | Data | MuST-C[3] | | |
|    | Sents | 250k | 264k | 274k |
| MT | Data | WMT14[4] | WMT13[5] | WMT14 |
|    | Sents | 4M | 14M | 36M |

Table 5: Statistics of data used for main experiments.

## B   Implementation Details

**Data**   We used MuST-C (Di Gangi et al., 2019) dataset with three language directions: English (En) to German (De), Spanish (Es), and French (Fr). For external MT data, we used WMT dataset with different years for each language pairs: WMT13 for

---

[3] https://mt.fbk.eu/must-c/
[4] https://www.statmt.org/wmt14/translation-task.html
[5] https://www.statmt.org/wmt13/translation-task.html

En-Es and WMT14 for En-De, En-Fr. For Open-Subtitle experiment, we used v2018[6] with 18M sentences for finetuning, while pretrained checkpoints are shared with WMT experiments.

**Models** We brought the model architecture from Ye et al. (2021) as strong baseline systems. Concretely, the model comprised of transformer encoder-decoder and modality-specific encoders. For speech inputs, we used 16-bit 16kHz raw signal as our audio input, and it is fed into the acoustic encoder, which is a sequence of Wav2Vec2 (Baevski et al., 2020) and convolutional subsampler. Due to the sequence length difference between speech and text inputs, the 2-layer convolutional subsamplers with kernel size 5 and stride 2 are used. For text inputs, conventional embedding layer with tokenized subwords are used. Output from the modality-specific encoders are subsequentially fed into the shared translation encoder-decoder.

Following recent works with competitive results (Ye et al., 2021, 2022; Han et al., 2021; Fang et al., 2022), we also leveraged pretraining strategies for speech and MT. While several previous works proposed various types of pretraining, we only exploited two types of pretraining: speech representation learning and MT. For speech, we used publicly available, Librispeech (Panayotov et al., 2015) trained Wav2Vec2 (Baevski et al., 2020) with base configuration[7]. For MT, we pretrained encoder-decoder on MT task with external dataset $\mathcal{D}_{\text{ext}}$. We used post-norm transformer-base (Vaswani et al., 2017) with shared embedding layer.

**Training & Evaluation** For training, unigram tokenizers are firstly trained using sentencepice (Kudo, 2018) with 10k joint vocabularies on the transcriptions and translation pairs from MuST-C. For main experiments, we use AdamW optimizer (Loshchilov and Hutter, 2017) with $\beta_1 = 0.9$ and $\beta_2 = 0.999$. Learning rate is $5 \times 10^{-5}$ with 25000 warmup steps and inverse square root scheduling. Weight for KD and CL was $\alpha_{KD} = 0.2$ and $\alpha_{CL} = 5.0$ For MT pretraining, we use AdamW optimizer ($\beta_1 = 0.9$, $\beta_2 = 0.98$) with learning rate $7 \times 10^{-4}$, 4000 warmup steps. Dropout ($p = 0.1$) and label smoothing ($p = 0.1$) is applied to both MT pretraining and finetuning.

Triplet-based joint training only with $\mathcal{D}_{\text{ST}}$ is considered as baseline, and various combinations of

training techniques are built based on this. All models are trained with fairseq[8] (Ott et al., 2019) using 4 Nvidia V100 GPUs. Following Liang et al. (2021), size of batches without CL is twice that of with CL for fair comparison. We averaged best 5 checkpoints based on ST BLEU score of MuST-C dev set. At evaluation, we used sacreBLEU[9] (Post, 2018). All models are trained with 3 random seeds and concatenated for statistical test through paired bootstrap resampling (Koehn, 2004).

## C  Relation between Continual Learning and Consistency Learning

In original EWC, Hessians is usually approximated as diagonals of Fisher information matrix $F$, as seen in EWC loss as follows:

$$\mathcal{L}_{EWC} = \sum_i F_{ii}(\theta_i - \theta_i^*)^2 \qquad (3)$$

where $\theta^*$ is the parameter obtained from previous tasks. In the context of ST, thanks to availability of MT data, we can remove diagonal approximation from EWC through well-known relation between KL divergence and Fisher information as

$$\mathcal{D}_{KL}(p_{\theta^*}||p_\theta) \approx E_z[\log p_{\theta^*} - \log p_\theta]$$
$$= \frac{1}{2}(\theta - \theta^*)^T F^*(\theta - \theta^*) \qquad (4)$$

where $F^*$ indicates that the Fisher is calculated at $\theta^*$.

Concretely, CL can be understood as regularizing the dropout submodel using full model's curvature, unlike EWC-like regularizations with the following differences: it uses online mini-batch throughout the training, not fixed subset of previous task's dataset (Kirkpatrick et al., 2017) or exponential moving average of mini-batch Fisher (Chaudhry et al., 2018); Fisher is computed at continuously varying parameter $\theta$, not fixed $\theta^*$. Despite these differences, the joint training strategy allows for the approximation of Eq 2 to be considered accurate, as the parameters will remain in the low-loss valley of MT throughout the training process. Intuitively speaking, dropout-averaged curvature information regularizes the gradient at each training step in a more knowledge-preserving manner, as opposed to simply summing multitask gradients.

---

[6]https://opus.nlpl.eu/OpenSubtitles-v2018.php
[7]https://dl.fbaipublicfiles.com/fairseq/wav2vec/wav2vec_small.pt

[8]https://github.com/facebookresearch/fairseq
[9]BLEU signature: nrefs:1|bs:10000|seed:12345|case:mixed|eff:no|tok:13a|smooth:exp|version:2.0.0

## D Comparison with joint pretraining (JPT) methods

Joint-pretraining (JPT) approaches have critical limitations in low-resource settings. JPT requires the preparation and aggregation of more data from multiple sources, as shown in Table 6. They also necessitate additional data augmentation, either at the phoneme level (STPT) or the unit level (SpeechUT). This, in turn, adds complexity to the training pipeline, such as the inclusion of a phoneme transcriptor (STPT) and a text-to-unit generator (SpeechUT), as illustrated below. In terms of computational efficiency, we also compare the GPU updates required by our approach with JPT. Our proposed contribution, which involves interleaving modality-specific models in terms of data and computation, leads to significant cost savings.

- STPT(Tang et al., 2022)
  - Joint pretraining: 16 A100, 12 gradient accumulation, 200k updates
  - Joint finetuning: 8 V100, 3 gradient accumulation, 50k updates

- SpeechUT(Zhang et al., 2022)
  - Text-to-unit (T2U) generator: not reported
  - Joint pretraining: 32 V100, 1 gradient accumulation, 400k updates
  - Task-specific finetuning: 8 V100, 4 gradient accumulation, 50k updates

- Ours
  - Joint finetuning: 4 V100, 4 gradient accumulation, 200k updates

## E Further analysis

**Forgotten MT knowledge cannot be restored** Advantage of adopting CL in ST can be understood as two-fold: the well-known regularization effect already discussed in original paper (Liang et al., 2021), and keeping MT knowledge for successful cross-modal transfer. To verify that BLEU improvements cannot be solely explained by the former, we tried to fix the model, which already had undergone catastrophic forgetting, and see whether it restores MT knowledge. Concretely, we initialize the whole model with final checkpoint from baseline of 2 and retrain it with KD+CL. As shown in Figure 3, it

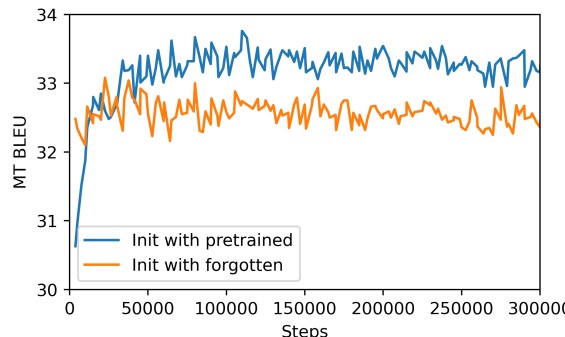

Figure 3: Evolution of MT BLEU with different parameter initialization.

is clear that forgotten MT knowledge cannot be restored even with CL. It also can be understood through our continual learning interpretation of CL discussed in Section C, that CL loss cannot provide credible curvature information outside of MT low-loss region.

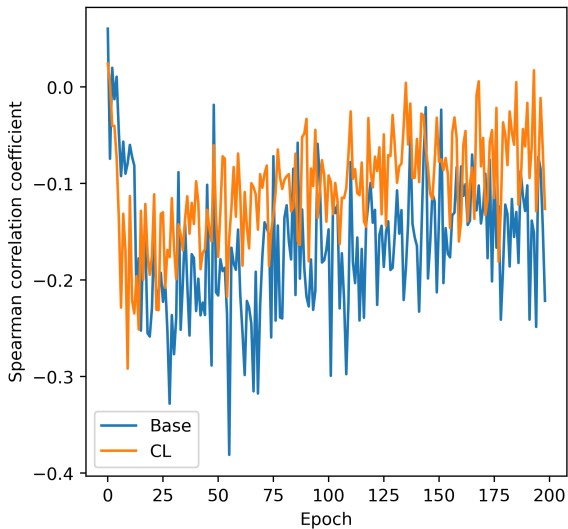

Figure 4: Epoch-wise Spearman correlation coefficient between batch order and their losses for IWSLT14 De-En. It is clear that CL alleviates imbalanced training problem.

**CL also remedies imbalanced training** Shao and Feng (2022) recently found that the problem of catastrophic forgetting arises in not only continual learning but also conventional static learning, especially in low-resource regime of complex tasks (e.g. MT). Concretely, they found the *imbalanced training* problem exists, that the models concentrate more on recently exposed data samples. While we established the relation between CL and catastrophic forgetting in cross-modal transfer, we fur-

| Models | Pretrainig Data | | |
|---|---|---|---|
| | Speech | ASR | MT |
| STPT(Tang et al., 2022) | 60k hours | 400 hours | 4.5M sentences |
| SpeechUT(Zhang et al., 2022) | 1.4k hours | 100 hours | 4.5M sentences |
| Ours | 960 hours | - | 4.5M sentences |

Table 6: Amount of pretraining data required for En-De

ther verify it with conventional MT through the lens of imbalanced training. We trained low-resource MT systems with IWSLT14 De-En with 160k sentences[10], and calculated epoch-wise Spearman correlation between order of training batches and their respective losses following Shao and Feng (2022), that negative coefficient implies imbalanced training. As Figure 4 shows, CL largely alleviates the problem of imbalanced training even without the use of additional teacher model originally proposed in Shao and Feng (2022). We also gathered epoch-wise normalized losses over the whole training and calculated Spearman correlation for quantitative comparison, and it reconfirms our findings ($r = -0.12$ for baseline vs. $r = -0.07$ for CL).

---

[10]https://iwslt.org/