# OpenReview forum: "Consistency is Key: On Data-Efficient Modality Transfer in Speech Translation"
_EMNLP/2023/Conference — EMNLP 2023 Findings_

### Official Review · Reviewer_8RCh · 2023-07-30

**Typos Grammar Style And Presentation Improvements:** 1. Clarifying the meaning of $\mathca…
**Soundness:** 3

**Excitement:**

3: Ambivalent: It has merits (e.g., it reports state-of-the-art results, the idea is nice), but there are key weaknesses (e.g., it describes incremental work), and it can significantly benefit from another round of revision. However, I won't object to accepting it if my co-reviewers champion it.

**Paper Topic And Main Contributions:**

During the fine-tuning stage of the progressive training for the speech translation task, previous work usually utilizes the machine translation (MT) task to help train the speech translation model. The analysis of this work shows that the MT task is not necessary after adding the consistency learning strategy. They adapt the R-drop method as the consistency learning and further improve the knowledge distillation (KD) method called cKD. Their experiments on MuST-C three datasets show the effect of their methods.

**Questions For The Authors:**

Some work (e.g., SATE, STEMM) has proved that KD is an effective way to bridge the modality gap. Why does it only achieve slight improvement in this work? What is the possible reason?

**Reasons To Accept:**

1. The analyses are based on strong baselines, and the Consistency Learning based on R-Drop shows a significant improvement.

2. The conclusions can benefit the Progressive training method for the speech translation task, making the whole training process more efficient.


**Reasons To Reject:**

1. The influence of the domain problem is not clearly claimed. I noticed that the XSTNet (Ye et al., 2021) shows that adding MT loss during the fine-tune stage is useful, which is inconsistent with your conclusions. The difference between the two papers is that they use in-domain MT data, while you use out-of-domain MT data. Thus, the influence of the domain problem has not been explored clearly, and whether it will affect your conclusion is unknown. This makes your conclusion not solid enough.
2. The contribution is limited. As shown in Table 2, joint pretraining methods do not rely on Progressive training. Furthermore, the proposed cKD method is not effective enough.
3. Since the motivation is to improve training efficiency, there is a lack of comparison about the training time between the baseline and proposed methods.


**Reproducibility:**

3: Could reproduce the results with some difficulty. The settings of parameters are underspecified or subjectively determined; the training/evaluation data are not widely available.

**Reviewer Confidence:**

4: Quite sure. I tried to check the important points carefully. It's unlikely, though conceivable, that I missed something that should affect my ratings.

---

> ### Author Rebuttal · Authors · 2023-08-29
>
> ### Regarding the domain problem
>
> Our experiments with in-domain MT are already presented in Table 4. With our proposed combination of KD and CL, progressive training does not bring about significant BLEU gains, which is consistent with out-domain experiments.. Moreover, it is evident that consistency plays a significant role in successful KD, as the inclusion of CL or cKD produces results that are comparable or even superior to the BLEU obtained from the incorporation of carefully selected, in-domain MT data (OpenSubtitles 28.91 vs. KD+CL 29.08 vs. cKD+CL 29.27). Our conclusion is solid, being already supported with both in- and out-of-domain MT data.
>
> ### Regarding JPT and training efficiency
>
> The catastrophic forgetting that occurs in cross-modal transfer is an important research topic in its own right. We have revealed that speech translation suffers from one, then demonstrated through extensive experimentation and analysis that CL can address this.
>
> In practical terms, as discussed in Appendix D, JPT methods necessitate a substantial amount of data collected from diverse sources and significant computational resources (e.g. 5x~10x GPU cost compared to ours). This implies that JPT methods cannot be generalized well for low-resource situations (e.g., low-resource languages). Our objective of achieving efficient cross-modal transfer without catastrophic forgetting provides additional low-resource and generalizable alternatives to the JPT approaches.
>
> ### Regarding the effectiveness of cKD
>
> We stress our gains are statistically significant for De and Es, derived from robust and thorough experiments. We employed bootstrap resampling with three random seeds to ensure scientifically meaningful conclusions (see Appendix B for details).
>
> ### Regarding the effectiveness of KD
>
> As KD itself does not improve speech translation a lot (JT-S-MT in table 2), for effective cross-modal KD, most of the previous non-JPT studies have utilized supplementary techniques to bridge the gap between modalities (e.g., contrastive learning, cross-modal mixup, or an additional adaptor module). In this work, we found that simple CL can be more effective than complex alternatives shown above (lines 218-221).
>
> Our method not only results in strong performance, but its simplicity also facilitates the seamless incorporation of the aforementioned “gap-bridging” methods. The investigation into its effectiveness remains a subject for future research.

---

### Official Review · Reviewer_muUy · 2023-08-05

**Soundness:** 3

**Excitement:**

3: Ambivalent: It has merits (e.g., it reports state-of-the-art results, the idea is nice), but there are key weaknesses (e.g., it describes incremental work), and it can significantly benefit from another round of revision. However, I won't object to accepting it if my co-reviewers champion it.

**Paper Topic And Main Contributions:**

The paper is focused on methods aiming to deal with the data scarcity issue in end-to-end speech translation (E2E ST). The authors investigated the effectiveness of progressive training, empirically found that it is not as effective due to learning-forgetting trade-off, and proposed a new method combining knowledge distillation and consistency learning to deal with catastrophic forgetting. The authors then propose consistency-informed KD (cKD) using token-level consistency to further improve the method.

**Reasons To Accept:**

The motivation of the work and of the proposed methods are clear.
The proposed approach combining KD and CL helps to deal with catastrophic forgetting and is more efficient than choosing the best D_ext, and is further improved using the cKD loss.

**Reasons To Reject:**

The claim that progressive training is ineffective could be better supported with more results from more language pairs (given that there are 8 available from MuST-C).

The authors also mentioned expensive computational overhead and made a point on training efficiency, thus it could be good to provide some self-contained quantitative comparisons.

It also seems that some results are difficult to reproduce due to MT checkpoints not being publicly available, if I'm not missing something here.

**Reproducibility:**

3: Could reproduce the results with some difficulty. The settings of parameters are underspecified or subjectively determined; the training/evaluation data are not widely available.

**Reviewer Confidence:**

2: Willing to defend my evaluation, but it is fairly likely that I missed some details, didn't understand some central points, or can't be sure about the novelty of the work.

**Typos Grammar Style And Presentation Improvements:**

Typos:
- l. 117 "trainscrption-translation" -> "transcription-translation"
- l. 563, "paris" -> "pairs"

Presentation:
- In Table 2, column De, the numbers are not well-aligned (ex. they could be centered around the dot)

---

> ### Author Rebuttal · Authors · 2023-08-29
>
> ### Q1
> Because our primary goal was a thorough reexamination of prior works, we focused on reporting results with statistical significance  for 3 language pairs. We are experimenting with 8 languages, but due to the resources required for the extensive experiments (24 models for each language direction), will report conclusive significance results to the camera-ready version.
>
> ### Q2
>  In Appendix D, we already presented a quantitative comparison with JPT methods, demonstrating that our method requires only about 1/10 to 1/5 of the GPU cost when compared to the JPT methods.
>
> ### Q3
> While our codebase and data are already reproducible being sourced from publicly available resources (e.g., fairseq and WMT), we will also provide an easier access to the data, code, and training scripts to ensure straightforward reproducibility.

---

### Official Review · Reviewer_zMRf · 2023-08-08

**Soundness:** 3

**Excitement:**

3: Ambivalent: It has merits (e.g., it reports state-of-the-art results, the idea is nice), but there are key weaknesses (e.g., it describes incremental work), and it can significantly benefit from another round of revision. However, I won't object to accepting it if my co-reviewers champion it.

**Paper Topic And Main Contributions:**

This paper investigates the effectiveness of progressive training, a commonly used technique for training end-to-end speech translation (ST) models by leveraging abundant machine translation (MT) data. The main contributions are:

1.	The authors re-evaluate progressive training and find it is actually ineffective for improving ST performance, despite its prevalence.
2.	They identify the core issue as the trade-off between learning ST and avoiding catastrophic forgetting of MT knowledge.
3.	To address this, they propose using consistency learning (CL) to alleviate forgetting, allowing better transfer from MT models.
4.	They show a simple combination of CL and knowledge distillation (KD) achieves strong results, outperforming progressive training.
5.	They further improve performance with a novel consistency-informed KD method that leverages token-level consistency information.

Overall, this paper makes important empirical and modeling contributions around effectively leveraging abundant MT data to improve low-resource ST in a data-efficient manner. The proposed CL and consistency-informed KD methods outperform previous techniques.


**Reasons To Accept:**

1.	Thoroughly evaluates and analyzes the common practice of progressive training for ST, showing it is actually ineffective despite its prevalence. This is an important empirical finding.
2.	Proposes a simple yet effective combination of consistency learning and knowledge distillation that achieves strong results without requiring extra training data. This provides a valuable data-efficient training approach.
3.	Introduces a novel consistency-informed knowledge distillation method that leverages intra-modal consistency for better inter-modal transfer. This is an interesting modeling contribution.


**Reasons To Reject:**

1.	Although the proposed method achieves remarkable improvements, the motivation behind introducing CL is unclear. The claim made in lines 164-173 is confusing and difficult for me to comprehend.
2.	Some arguments lack supporting evidence. For instance, the authors state in line 131 that "The external MT diverts the model's focus from ST," but there is no analysis provided regarding the interference between MT and ST.
3.	The improvements of the cKD method are limited.
4.	CL is indeed an effective regularization method that helps alleviate overfitting. However, it is necessary to demonstrate that the improvements in the MT model are not solely due to regularization but rather to the avoidance of forgetting. Additionally, the significant improvements achieved by using the CL method alone, as shown in Table 4, suggest that the KD technique may not be as important. The better regularization method may be crucial for ST fine-tuning.


**Reproducibility:**

4: Could mostly reproduce the results, but there may be some variation because of sample variance or minor variations in their interpretation of the protocol or method.

**Reviewer Confidence:**

4: Quite sure. I tried to check the important points carefully. It's unlikely, though conceivable, that I missed something that should affect my ratings.

---

> ### Author Rebuttal · Authors · 2023-08-29
>
> ### Q1 & Q2
> We argue that these two points can be addressed through rewriting. Due to the space limitations of a short paper, we have included many supporting pieces of evidence in the Appendix (C for motivation of CL and E for additional analysis). These can be incorporated into the main text to better motivate CL and support claims with evidence.
>
> Regarding MT-ST interference, it has already been reported in prior works (e.g., section 5.2 of https://arxiv.org/abs/2204.05409). We reconfirmed this in our preliminary experiment, where a significant amount of additional MT data exhibited poor performance and, in some cases, even failed to converge. We will supplement the discussion about it.
>
> ### Q3
> We stress our gains are statistically significant for De and Es, derived from robust and thorough experiments. We employed bootstrap resampling with three random seeds to ensure scientifically meaningful conclusions (see Appendix B for details).
>
> ### Q4
> Appendix E supports the improvement is from “avoidance of forgetting”, as forgotten MT knowledge cannot be restored even with CL. It also can be understood through our continual learning interpretation of CL in Appendix C that CL loss cannot provide credible curvature information outside of MT low-loss region.
>
> Regarding the strong results of CL-only in Table 4, these were observed only when in-domain MT data is available (as shown in Table 3). Also, note that even in the case of in-domain MT, cKD outperforms both CL and KD+CL, indicating the importance of intra-modal consistency for a successful KD.

---

### Official Review · Reviewer_mLFp · 2023-08-10

**Soundness:** 3

**Excitement:**

3: Ambivalent: It has merits (e.g., it reports state-of-the-art results, the idea is nice), but there are key weaknesses (e.g., it describes incremental work), and it can significantly benefit from another round of revision. However, I won't object to accepting it if my co-reviewers champion it.

**Paper Topic And Main Contributions:**

The paper proposed a method which combines knowledge distillation consistency learning to improve ST. This paper identifies learning-forgetting trade-off as a critical obstacle, then hypothesize and verify that consistency learning (CL) breaks the dilemma of learning-forgetting.

**Reasons To Accept:**

1 This paper conducts an empirical investigation and finds that progressive training is ineffective and proposes a more effective method.

2 This paper proposes cKD to dynamically utilize intra-modal consistency for cross-modal KD, which is the major novelty of this paper.


**Reasons To Reject:**

The paper lacks sufficient novelty.

**Reproducibility:**

3: Could reproduce the results with some difficulty. The settings of parameters are underspecified or subjectively determined; the training/evaluation data are not widely available.

**Reviewer Confidence:**

3: Pretty sure, but there's a chance I missed something. Although I have a good feel for this area in general, I did not carefully check the paper's details, e.g., the math, experimental design, or novelty.

---

> ### Author Rebuttal · Authors · 2023-08-29
>
> ### Q1
> Our novelty is, as the reviewer also summarized, a thorough reinvestigation of the commonly held belief in the effectiveness of progressive training, and proposing intra-modal consistency to augment KD.
>
> We believe the extensive evaluation we conducted across a wide search space, coupled with robust statistical testing is a significant contribution and also could not find an existing finding similar to this, which was encouraged to be mentioned in the review when arguing the lack of novelty, per reviewing policy.

---

### Meta-Review · Area_Chair_gd1u · 2023-09-14

**Recommendation:** 3

**Metareview:**

This paper shows the limitations in progressive training for speech translation (where external text translation data is used during fine-tuning), and then proposes an approach to address this shortcoming. The reviewers agreed that the experiments were sound in showing the problems associated with progressive training as well as the improvements that can be achieved by the new approach. However, the reviewers also had concerns that further analyses could have improved some of the claims in the paper. Some reviewers also had concerns about the novelty of the work, leading to overall ambivalent excitement. Nevertheless, the overall soundness is good.

---

### Decision · Program_Chairs · 2023-10-07

**Decision:**

Accept-Findings

**Comment:**

This paper shows the limitations in progressive training for speech translation (where external text translation data is used during fine-tuning), and then proposes an approach to address this shortcoming. The reviewers agreed that the experiments were sound in showing the problems associated with progressive training as well as the improvements that can be achieved by the new approach. However, the reviewers also had concerns that further analyses could have improved some of the claims in the paper. Some reviewers also had concerns about the novelty of the work, leading to overall ambivalent excitement. Nevertheless, the overall soundness is good.